# Multifaceted Functional Liposomes: Theranostic Potential of Liposomal Indocyanine Green and Doxorubicin for Enhanced Anticancer Efficacy and Imaging

**DOI:** 10.3390/pharmaceutics17030344

**Published:** 2025-03-07

**Authors:** Wei-Ting Liao, Dao-Ming Chang, Meng-Xian Lin, Te-Sen Chou, Yi-Chung Tung, Jong-Kai Hsiao

**Affiliations:** 1Department of Medical Imaging, Taipei Tzu Chi General Hospital, Buddhist Tzu-Chi Medical Foundation, New Taipei City 23142, Taiwan; r05b42035@ntu.edu.tw (W.-T.L.); tch36363@tzuchi.com.tw (M.-X.L.); zackmiku0523@gmail.com (T.-S.C.); 2School of Medicine, Tzu Chi University, Hualien 97004, Taiwan; 3Research Center for Applied Sciences, Academia Sinica, Taipei 11529, Taiwan; dmchang@gate.sinica.edu.tw

**Keywords:** ICG, doxorubicin, liposome, photothermal effect, photodynamic therapy, ferroptosis

## Abstract

**Background/Objectives:** Liposomal drug formulations improve anticancer treatment efficacy and reduce toxicity by altering pharmacokinetics and biodistribution. Indocyanine Green (ICG), an FDA-approved near-infrared imaging agent, exhibits photosensitivity, photothermal effects, and potential ferroptosis induction, enhancing anticancer activity. Doxorubicin (DOX), widely used for treating breast, ovarian, and liver cancers, is limited by cardiotoxicity, requiring dosage control. Incorporating ICG and DOX into liposomes enables medical imaging, controlled drug release, reduced administration frequency, and fewer side effects. This study aims to develop liposomes encapsulating both ICG and DOX and evaluate their theranostic potential in in vitro and in vivo lung adenocarcinoma models. **Methods:** Liposomes containing ICG and DOX (Lipo-ICG/DOX) were synthesized using an active loading method and characterized for size (~140 nm), lipid, and drug concentrations. In vitro studies using A549 lung cancer cells assessed liposome uptake via fluorescence microscopy, while in vivo xenograft models evaluated therapeutic efficacy. **Results:** Lipo-ICG/DOX showed uptake in A549 cells, with ICG localizing in lysosomes and DOX in nuclei. Treatment reduced cell viability significantly by day three. In vivo imaging demonstrated the retention of liposomes in tumor sites, with ICG signals observed in the liver and intestines, indicating metabolic routes. When combined with 780 nm light exposure, liposomes slowed tumor growth over 12 days. Mechanistic studies revealed combined ferroptosis and apoptosis induction. **Conclusions:** Lipo-ICG/DOX demonstrates strong theranostic potential, integrating imaging and therapy for lung adenocarcinoma. This multifunctional formulation offers a promising strategy for improving cancer treatment efficacy while minimizing side effects.

## 1. Introduction

Cancer remains one of the leading causes of mortality worldwide, presenting significant challenges to global health systems. Theranostics, which integrates diagnostic and therapeutic approaches, has emerged as a transformative tool in cancer treatment. By enabling precise tumor detection and delivering targeted therapies, theranostics enhances treatment efficacy, reduces side effects, and facilitates real-time monitoring of therapeutic outcomes, paving the way for more personalized and effective strategies against this complex disease. Furthermore, multifaceted treatment combining multiple therapeutic strategies, such as chemotherapy and photothermal therapy, has demonstrated even greater potential in cancer treatment. Therefore, the development of multifaceted therapies for theranostics holds great promise for improving cancer treatment, making it more precise, effective, and tailored to individual patient needs.

Indocyanine Green (ICG) is a cyanine dye that has been widely used in medical diagnostics for over 60 years. Clinically, ICG is employed in angiography and liver function evaluation through the ICG test [1]. Beyond its established role as a safe and non-toxic contrast agent, ICG has recently gained attention for its therapeutic applications. When taken up by cells and exposed to near-infrared light, ICG induces localized temperature increases and generates reactive oxygen species (ROS), leading to apoptosis and enabling its use in photodynamic therapy for cancer treatment [2]. Furthermore, emerging studies suggest that ICG exhibits anticancer properties associated with ferroptosis, expanding its potential in oncology [3].

ICG is a stable amphipathic compound that can be dissolved in both aqueous and oil phases, and it can withstand high temperatures and maintain its integrity across a pH range of 6.0–8.0 [4]. In the bloodstream, ICG binds with human serum albumin (HSA) to form a stable complex, which shifts its maximum optical absorption peak from approximately 780 nm to 800 nm. This shift, coupled with the superior tissue penetration of longer wavelengths, makes ICG a highly effective agent for tumor phototherapy [5,6]. Despite challenges in storing ICG in an aqueous solution for extended periods, encapsulation in liposomes has been employed to prolong its biological lifetime. Liposomes not only extend ICG stability but also facilitate controlled drug release [7,8]. Coating the liposome surface with polyethylene glycol (PEG) further improves storage capacity and stabilizes ICG within the bilayer phospholipid structure, providing a robust platform for biomedical applications [9,10,11]. Encapsulating ICG within a double-layered phospholipid structure provides better photostability and results in a high permeability and long retention effect in animal models [12].

Doxorubicin is a bright red antibiotic with potent antitumor properties, proven effective in treating lymphomas and acute leukemia. It works by intercalating into DNA, forming complexes that inhibit DNA replication and block the activity of DNA topoisomerase II beta, ultimately inducing cell apoptosis [13]. However, a significant limitation of doxorubicin is its potential to cause irreversible severe heart failure when the cumulative dose exceeds 550 mg/m^2^, restricting its use in cancer therapy. To address this issue, Doxil, a liposomal nanoformulation of doxorubicin, was developed. This encapsulated form offers a vastly different pharmacokinetic profile compared to conventional doxorubicin chemotherapy, improving therapeutic efficacy while reducing cardiotoxicity and other side effects [14].

The growing recognition of the benefits associated with the combination of ICG and DOX as both therapeutic and theranostic agents for liposomal applications has spurred the development of a range of synthetic strategies and treatment methodologies centered on Lipo-ICG/DOX [15,16,17,18,19]. Research has revealed the photothermal-triggered enhancement of drug release using an 808 nm laser directed at DOX/ICG liposomal nanogels [19]. Additional advancements have also been made by integrating polydopamine, ICG, and DOX in methanol, followed by filtration through a 0.45 μm filter, culminating in the creation of an ICG/DOX nanoformulation with a diameter of 285.4 nm. The theranostic efficacy of this formulation against MDA-MB-231 xenograft models has been empirically confirmed [17].

For the synthesis and exploration of ICG- and DOX-incorporated liposomes in cancer treatment, several attempts have been made across different studies. For instance, Wang et al. (2023) employed an active loading strategy with an (NH4)_2_SO_4_ gradient to encapsulate DOX, subsequently attaching a DOPE-ICG complex on the liposome surface and relying on sonodynamic therapy (SDT) [15]. In contrast, Xue et al. (2018) used reverse-phase evaporation to form ICG/DOX liposomes for HepG2 cells, highlighting how 808 nm laser irradiation promotes DOX release through photothermal effects [16]. Similarly, Yu et al. (2018) incorporated hydrogel precursors into ICG-mediated photothermal liposome systems targeting 4T1 breast cancer [19], while Lu et al. (2023) co-loaded DOX and ICG in methanol within a polydopamine–liposome framework for MDA-MB-231 xenografts [17]. Despite these variations in synthesis protocols, tumor models, and dosages, all studies underscore the synergistic anticancer benefits of combining ICG and DOX within a single liposomal or nanoparticle platform—whether by photothermal or sonodynamic mechanisms.

In this study, we developed liposomes encapsulating both ICG and doxorubicin (Lipo-ICG/DOX) to enhance anticancer efficacy with theranostic potential. The ICG loaded in the liposomes has several functions for cancer treatment. First, ICG can be exploited as an imaging agent for medical diagnostics. Furthermore, ICG itself can be used for anticancer therapy by inducing apoptosis and ferroptosis in cancer cells, especially when ICG is irradiated by light. The therapeutic effects and the related mechanisms have been discussed in previous papers [2,20]. Furthermore, ICG makes the liposomes temperature-sensitive liposomes, enabling controlled drug release through near-infrared light-induced heating [21]. In addition, the liposome provides improved biodistribution and bioavailability, leading to an increased therapeutic index. The PEG coating further enhances the storage stability, hemodynamic performance, and fluorescence stability of the liposome [19,22]. While many nanomedicine formulations are clinically available, their high costs remain a significant barrier compared to traditional medications [23]. Fabricating the liposomes using cost-effective synthesis methods could significantly benefit patients by reducing expenses. The primary objective of this study is to fabricate ICG- and DOX-incorporated liposomes to investigate their theranostic potential both in vitro and in vivo. A secondary goal is to validate the anticancer efficacy of the synthesized nanoformulation.

## 2. Materials and Methods

### 2.1. Preparation and Characterization of ICG/DOX Liposomes

In the experiments, we synthesized the Lipo-ICG/DOX through an active loading method in this study. In order to fabricate the liposomes, we prepared the lipid by mixing DPPC/cholesterol/DSPE-PEG2000 (DPPC, cholesterol, DSPE-PEG2000 were purchased form Avanti Polar Lipids Inc., Alabaster, AL, USA) at a ratio of 3:1:1 (*w*/*w*/*w*) and dissolved the ICG powder (Daiichi Sankyo Propharma Co., Ltd., Takatsuki Plant., Takatsuki-shi, Osaka, Japan) in chloroform (288306, Sigma-Aldrich, Burlington, MA, USA) with a concentration of 8 mg/mL. For every 1 mL liposome, we used a total weight of 10 mg lipid and 0.5 mg of ICG. After dehydration using a rotary evaporator at 40 °C, a clear green film could be obtained.

To encapsulate doxorubicin within the liposomes, the clear green lipid film was dissolved in an equal volume of a 120 μM ammonium sulfate solution (pH = 6) (A4418, Sigma-Aldrich, Burlington, MA, USA) at a concentration of 10 mg lipid per 1 mL of liposome product. The solution was sonicated at 50 °C for 20 min, transferred to a syringe, and filtered through a 200 nm pore membrane (WHA111106, Whatman Nuclepore Track-Etched Membrane, Sigma-Aldrich, Inc., St. Louis, MO, USA) at 60 °C to obtain ICG liposomes with a diameter of less than 200 nm. The fabricated liposomes were then dialyzed using 8 kDa pre-wetted dialysis tubing (132580, Thermo Fisher Scientific, Waltham, MA, USA) in the dark at 4 °C for four days, with the water exchanged daily. This process established an ammonium sulfate concentration gradient inside the liposomes. Subsequently, the liposome solution was mixed with an aqueous solution of doxorubicin (Adriamycin Inj. 2 mg/mL, Pfizer, Bentley, Western Australia, Australia) and heated at 60 °C for over 45 min, allowing the replacement of ammonium sulfate with doxorubicin [24]. The final pH of the fabricated liposome solution is 7.

The properties of the fabricated ICG/DOX liposomes were analyzed as described previously [20]. Specifically, lipid concentration was quantified using the Stewart assay, and liposome size distribution was characterized through dynamic light scattering (DLS) measurements (Brookhaven 90 Plus Nanoparticle Size Analyzer, Holtsville, NY, USA). In addition, the concentrations of ICG and doxorubicin were estimated based on optical absorbance measurements using a spectrophotometer.

### 2.2. Cell Culture

For the cell experiments, the human lung adenocarcinoma cell line A549 was obtained from the Bioresource Collection and Research Center (BCRC, Hsinchu, Taiwan). The cells were cultured in Dulbecco’s Modified Eagle Medium (DMEM) (Gibco 11965, Thermo Fisher Scientific), supplemented with 10% fetal bovine serum (FBS) (Gibco 10082, Thermo Fisher Scientific). A total of 5 × 10^4^ A549 cells were maintained in a T25 flask (156367, Thermo Fisher Scientific). The cells were incubated at 37 °C with 5% CO_2_ until they reached confluence, after which subsequent experiments were performed.

### 2.3. Cellular Uptake of Liposomes by A549 Cells

To evaluate the cellular uptake of Lipo-ICG/DOX, 2 × 10^4^ A549 cells were seeded onto an 8-well cell culture slide (30108, SPL, Pocheon, Republic of Korea) and cultured for one day in a 37 °C incubator with 5% CO_2_ to allow for attachment. Subsequently, prepared lipo-ICG/DOX in the complete medium was added to the cells. The A549 cells were treated with Lipo-ICG/DOX at two concentrations: ICG 31 μg/mL with doxorubicin 25 μg/mL and ICG 62 μg/mL with doxorubicin 50 μg/mL. After a four-hour treatment, the liposomes were washed off with DPBS (14190-144, Gibco), and a mixture of LysoTracker (L7528, LysoTracker Red DND-99, Thermo Fisher Scientific, Waltham, MA, USA) and serum-free DMEM was added at a working concentration of 75 nM. Following an hour of incubation, the LysoTracker was washed off using DPBS, and the cells were fixed with 4% paraformaldehyde (PFA). The nuclei were then stained with DAPI (D1306, Invitrogen, Thermo Fisher Scientific, Waltham, MA, USA). The slide was gently rinsed with DPBS and mounted using the mounting medium (H-1400-10, Vector Laboratories, Newark, CA, USA).

### 2.4. Photothermal Effect Validation

The photothermal effect of the Lipo-ICG/DOX was validated using a thermal imaging camera (Ti55, IR FlexCam Thermal Imager, Fluke Corporation, Washington, DC, USA). For the experiment, 0.5 mL of the concentrated liposomes containing ICG (203 μg/mL) and doxorubicin (106 μg/mL) was placed on a custom-made acrylic rack and irradiated with 780 nm light at a power of 800 mW (M780LP1, Thorlabs, Inc., Newton, NJ, USA) for 0 to 20 min. The temperature changes were continuously monitored and recorded every minute for 20 min using the thermal imaging camera.

### 2.5. Photothermal Effect-Induced Drug Release

To further demonstrate light-controlled drug release from the Lipo-ICG/DOX liposomes, 0.5 mL of the concentrated liposomes was placed on a custom-made acrylic rack and irradiated with 780 nm light at a power of 800 mW. Following irradiation, the liposomes were centrifuged using a microcentrifuge tube (UFC510024, 100 kDa, MERCK Amicon^®^ Ultra-0.5, Darmstadt, Germany) with a 100 kDa filter. During centrifugation, the irradiated liposome solution was placed in the inner tube, and the entire setup was subjected to 4000 G, 4 °C, for at least 15 min. After centrifugation, the intact liposomes remained in the inner tube, while the solution containing the released drugs was collected from the outer tube. By measuring the drug concentration in the outer tube, the light-controlled drug release under specific light doses can be quantified.

### 2.6. In Vitro Cell Viability Analysis

To evaluate drug efficacy, in vitro cell experiments were conducted using the A549 lung cancer cell line. The cells derived from the same batch were divided into five populations and treated with liposomes at various concentrations (ICG: 1.25 to 10 μg/mL, doxorubicin: 0 to 62.5 nM). The A549 cells were cultured in 96-well plates with 2.5 × 10⁴ cells per well and incubated in a cell culture incubator maintained at 37 °C with 5% CO_2_ for 1 to 3 days. Following the treatment, the medium containing liposomes was removed, and the cells were washed with DPBS before adding DMEM supplemented with the PrestoBlue™ Cell Viability Reagent (Invitrogen A13262, Thermo Fisher Scientific) to estimate cell viability for drug efficacy quantification. Fluorescence intensity was measured at an excitation/emission wavelength of 560/590 nm using a multi-mode microplate reader (BioTek Synergy 2, Thermo Fisher Scientific) in bottom-read mode.

### 2.7. RNA Sequencing (RNA-Seq) Analysis

A549 cells were cultured in T75 flasks (156499, Nunc™ EasYFlask™ Cell Culture Flasks, Thermo Scientific™, Waltham, MA, USA) and divided into three treatment groups: (1) treated with empty liposomes in complete DMEM for 4 h; (2) treated with Lipo-ICG/DOX in complete DMEM at concentrations of ICG (50 μg/mL) and DOX (38.4 μg/mL) for 4 h, followed by washing with DPBS; and (3) treated with Lipo-ICG/DOX in complete DMEM (ICG: 50 μg/mL, DOX: 38.4 μg/mL) for 4 h, followed by washing with DPBS and irradiation with 780 nm light for 20 min. All groups were then incubated at 37 °C with 5% CO_2_ for 24 h to recover from drug and light treatment. Cells were subsequently harvested using 3 mL of RNA extractor (DPT-BD2 Smart RNA Extractor, TOOLS, New Taipei City, Taiwan). Total RNA was extracted using a silica-based column method, and RNA quality was assessed with a bioanalyzer system (2100, Agilent Technologies, Waldbronn, Germany). Libraries were constructed using 1000 ng of ribosomal RNA-depleted RNA using the library preparation kit (TruSeq Stranded Total RNA with Ribo-Zero Plant, Illumina, San Diego, CA, USA) following the manufacturer’s protocol. Sequencing was performed on a sequencing instrument (NovaSeq 6000, Illumina, San Diego, CA, USA) with a 2 × 150 bp paired-end protocol. FASTQ reads were generated using Illumina’s bcl2fastq software v2.20, and adapter removal and quality trimming were performed using Trimmomatic. The RNA sequencing data were aligned to the GRCh38 human reference genome using the HISAT2 aligner. Differential expression analysis was conducted with StringTie and DESeq2, incorporating genome bias detection and correction. Functional enrichment analysis of differentially expressed genes across the three treatment groups was performed using clusterProfiler.

### 2.8. In Vivo Experiments

To assess the anticancer efficacy of the liposomes, animal experiments were conducted in this study. All animal experiments were performed in accordance with the National Institutes of Health (NIH) guidelines for the care and use of laboratory animals. The study protocol (109-IACUC-014) was approved by the Institutional Animal Care and Use Committee. Female BALB/cAnN.Cg-Foxnlnu/CrlNarl nude mice, aged 6–8 weeks, were used for the experiments. A549 cells (1 × 10^6^) were implanted into both flanks of each mouse. Tumor size was measured every other day using calipers to record the length (*L*) and width (*W*), and the tumor volume (*V*) was calculated using *V* = 0.5 × *L* × *W*^2^. Treatment with liposomes and light irradiation began when the tumor volumes reached 45–50 mm^3^. The mice were divided into three groups, each consisting of four animals. One group received no treatment as the control group. Two treatment groups received liposomes containing ICG (651 μg/mL) and doxorubicin (520 μg/mL) in 200 μL, administered via intraperitoneal injection every other day. One of the treatment groups also received 780 nm light irradiation for 15 min daily using a near-infrared laser (High Power LED driver, THORLABS, Münchner, Germany). For irradiation, the laser light source was positioned 1 cm above the xenograft, delivering a fluence rate of 1 W/cm^2^.

### 2.9. In Vivo and Ex Vivo Imaging

After 14 days of treatment with Lipo-ICG/DOX and light irradiation, the nude mice were anesthetized with isoflurane and imaged using an IVIS50 imaging system (Xenogen, PerkinElmer, Waltham, MA, USA) at 3, 6, 24, and 30 h post-treatment. Whole-body imaging, including the transplanted tumor, was performed using an excitation wavelength of 780 nm, with fluorescence captured through an 845 nm filter. After the imaging, the mice were euthanized, and their organs—including the heart, lungs, liver, kidneys, spleen, gastrointestinal (GI) tract, and tumor xenograft—were collected for additional IVIS imaging. The tumor volumes across the three groups and the fluorescent signal intensities in the organs, as measured by IVIS, were analyzed using Student’s *t*-test. A *p*-value of <0.05 was considered statistically significant. The detailed liposome drug concentrations used in different in vitro and in vivo experiments are listed in the Appendix A.

## 3. Results

### 3.1. Liposomal Encapsulation of ICG and DOX

#### 3.1.1. Liposome Properties

The DLS analysis revealed that the particle sizes were uniformly distributed both with and without ICG and doxorubicin encapsulation. We observed precipitation between ICG and DOX, leading us to adopt the active loading method. After encapsulating ICG and doxorubicin, the average particle size was approximately 140 nm with the PDI value of 0.107 and an estimated concentration of 3.5 × 10^12^ liposomes per mL (Figure 1A). The liposomes were concentrated using a centrifuge to achieve ICG and doxorubicin concentrations of 203.3 μg/mL and 105.6 μg/mL, respectively. The concentrations were estimated by the absorbance spectra of ICG and doxorubicin dissolved in 80% methanol (Figure 1B,C). The drug encapsulation efficiency was calculated by dividing the amount of the drug encapsulated by the total amount used during synthesis based on optical absorbance values. The encapsulation efficiencies were estimated to be 67.1% for ICG and 52.2% for doxorubicin based on the calibration curves established using the absorbance measurement shown as Appendix A. The lipid concentration, measured using the Stewart assay, was found to be 1297.6 μg/mL. The fabricated liposomes were subsequently used in cell and animal experiments.

#### 3.1.2. Photothermal Effect and Controlled Drug Release

In order to demonstrate the photothermal effect of the Lipo-ICG/DOX, the liposomes were irradiated using light with the center wavelength of 780 nm. The 0.2 mL liposomes were placed in a 0.5 mL Eppendorf on a self-made small stand and photographed for 20 min with a thermal imaging camera in a time-lapse manner. The results show that the temperature of the Lipo-ICG/DOX rose by about 20 °C after 20-min irradiation (Figure 2A). Furthermore, irradiation can also be used to control the drug release of the liposomes. In the experiments, we irradiated the Lipo-ICG/DOX, and the results are shown in Figure 2B; the ICG and doxorubicin were quickly released within 5 min estimated using the microcentrifuge tubes and light absorbance measurement. We expect that the liposomal phospholipid membrane will become unstable after light exposure due to the ICG-induced temperature increase. Therefore, the controlled release of the encapsulated drug from the liposomes can be observed.

### 3.2. In Vitro Model

#### 3.2.1. Lipo-ICG/DOX on A549 Cell Line

To study the effects of the liposomes on in vitro models, A549 cells were treated with Lipo-ICG/DOX with two different concentrations for four hours. The medium was removed, and the cells were washed with DPBS and subsequently fixed and stained. The stained cells were then visualized using an inverted fluorescence microscope, as shown in Figure 3A. The images show the co-localization of the ICG and LysoTracker signals, indicating that ICG predominantly accumulated in the lysosomes (Figure 3B). In contrast, the overlapping DAPI and doxorubicin signals suggested that doxorubicin localized to the nucleus. Additionally, it was observed that the higher doxorubicin concentration caused the DAPI staining to appear more diffuse, whereas the lower doxorubicin concentration had less effect on nuclear staining (Figure 3C). This phenomenon may be attributed to the anticancer mechanism of doxorubicin, wherein it binds to DNA and inhibits nucleic acid synthesis and potentially interferes with effective nuclear staining.

#### 3.2.2. Treatment Efficacy

To estimate the anticancer efficacy of the liposomes, A549 cells are exploited to test the cell viability under the liposome treatment. In the test, we treated the cells with the liposomes at five concentrations for 23, 48, and 72 h, and the results are plotted in Figure 4. The treatment efficacy of combinations of the free drugs on the cells can also be compared, as shown in Appendix A. It was found that the liposomes containing doxorubicin can effectively reduce the activity of cancer cells in three days. In contrast to doxorubicin, ICG itself has minimal effects on cell viability, suggesting its great cell compatibility.

#### 3.2.3. RNA-Seq Analysis

Figure 5 shows the results of the RNA-seq performed on the A549 with different treatments. Our RNA-seq analysis identified several key pathways linked to apoptosis, ferroptosis, and cell cycle regulation, highlighting the multifaceted anticancer mechanisms of Lipo-ICG/DOX on A549 cells. For genes related to the apoptosis pathway (Figure 5A), LMNB1 exhibited the highest expression in the control group and the lowest in the lipo-ICG/DOX group treated with 780 nm light, consistent with its role in cell development suppression upon downregulation. Similarly, BIRC5, an inhibitor of apoptosis that promotes cell division and progression, and AKT1, which suppresses apoptosis via the IL-3 signaling pathway, were both highly expressed in the control group. In contrast, BIRC3, another IAP family member known to resist apoptosis in cancer cells, displayed increased expression following treatment. Pro-apoptotic genes such as DDIT3, an inhibitor of CCAAT/enhancer-binding protein (C/EBP) function, and GADD45A and GADD45B, which are involved in apoptosis induction, were upregulated in the lipo-ICG/DOX-treated groups, likely due to DOX’s DNA-damaging effects. Additionally, CTSV, which plays a role in apoptosis and lung cancer metastasis, and PMAIP1, a key pro-apoptotic regulator, showed significantly elevated expression in the treatment group. These findings indicate that lipo-ICG/DOX treatment, particularly under 780 nm light activation, promotes apoptosis and pro-apoptotic activity, whereas the control group exhibits stronger anti-apoptotic signaling.

In the ferroptosis pathway, LMNB1 downregulation reduces iron levels, while its upregulation induces mitochondrial damage and ferroptosis. Fatostatin, an inhibitor of sterol regulatory element-binding protein (SREBP), accelerates ferroptosis by inhibiting the AKT/mTORC1/GPX4 pathway, whereas AKT1 upregulation induces GPX4, reducing ferroptosis. Additionally, ATF4 inhibits ferroptosis by upregulating DDIT3. In our RNA-seq data, ATF4 expression was the lowest in the lipo-ICG/DOX with 780 nm light treatment group, indicating the highest level of ferroptosis in this condition.

In the cell cycle pathway (Figure 5B), GADD45A and GADD45B regulate both apoptosis and cell proliferation, with their elevated expression inhibiting proliferation and slowing growth in the G2/M phase. Conversely, E2F2, which facilitates S-phase entry, was downregulated in the treatment group. Similarly, PLK1, a key regulator of mitotic entry and a marker of heightened cell proliferation, was overexpressed in the control group. Additionally, CDC20 and CDC45, which are associated with aberrant proliferation and poor prognosis, showed reduced expression in the lipo-ICG/DOX with 780 nm light treatment group. These findings suggest that the treatment effectively suppresses key regulators of cell cycle progression, contributing to its antiproliferative effects.

In both the immunofluorescence (Figure 5C) and RNA-seq TPM data (Figure 5D), GPX4 expression, an essential marker of ferroptosis, was higher in the control group but decreased following treatment with lipo-ICG/DOX and 20-min 780 nm light exposure. Since GPX4 inhibition leads to lipid peroxidation, this reduction indicates an increase in ferroptosis. Figure 5C presents in vivo data from A549 cell-derived tumors injected into the flanks of nude mice. Once the tumors reached approximately 50 mm^3^, they were treated with the Lipo-ICG/DOX and exposed to 780 nm light. After 12 days, the mice were sacrificed, and tumor sections were analyzed for GPX4 fluorescence staining. Because these sections were embedded in paraffin, the deparaffinization process removed the lipo-ICG/DOX, preventing the detection of ICG and DOX signals via fluorescence microscopy. To determine whether 780 nm light could induce phototherapy effects and ferroptosis within the tumor, we captured images from both the tumor edge and interior, revealing similar GPX4 expression levels in both regions. Since RNA-seq data from Figure 5A,B did not specifically highlight GPX4, we performed fluorescence staining to assess its expression in vivo. To further confirm these findings, we also included in vitro GPX4 expression data in Figure 5D. Together, these results demonstrate that lipo-ICG/DOX treatment combined with 780 nm light exposure effectively induces ferroptosis in both in vitro and in vivo models.

### 3.3. In Vivo Model

#### IVIS Image of Lipo-ICG/DOX in Nude Mouse

In order to observe the biodistribution of the liposomes in the animals, IVIS imaging was performed on the nude mice treated with the Lipo-ICG/DOX. The images showed that the tumor exhibited the highest near-infrared signal 3–6 h after the Lipo-ICG/DOX administration and still retained the signal intensity 30 h after the administration (Figure 6A). Ex vivo IVIS imaging of organs and tumor xenograft 30 h after the liposome administration showed that most of the ICG signals are in the kidneys, liver, and the GI tract. In addition, the images showed that the tumor xenograft exhibited near-infrared signals that could still be traced (Figure 6B). We further compared the Lipo-ICG/DOX treatment group to the control group using a bar chart and found the infrared signal of the heart, the target organ of doxorubicin cardiotoxicity, is only minimally elevated, whereas the tumor signal intensity is elevated significantly (Figure 6C). Figure 6C represents the average radiant efficiency normalized by the area. As the ex vivo IVIS was performed 30 h after lipo-ICG/DOX delivery, it represents both the metabolism and accumulation of lipo-ICG/DOX in each organ. The total signal intensity of the near-infrared was illustrated by the pie chart and revealed more than 80% of the signals from the GI tract (Figure 6D). The total radiant efficiency reflects the total amount of photons passing through. As a result, the digestive system, having the largest area, appears to have the highest signal. Using the total radiant efficiency, we analyzed the final distribution of lipo-ICG/DOX 30 h after injection, demonstrating the metabolic pathway from the liver to the bile and, ultimately, excretion from the digestive system.

We observed that tumor xenografts grew rapidly following the implantation in both the control group and the only Lipo-ICG/DOX-treated group. However, this rapid growth was not observed in the mice treated with the Lipo-ICG/DOX and light radiation. The difference in the tumor growth was statistically different between the control group and the Lipo-ICG/DOX with the light irradiation group on days 3 and 12 (Figure 7).

## 4. Discussion

Our initial synthesis method involved mixing ICG with lipids in chloroform. After vacuum rotation evaporation, the resulting lipid film was mixed and shaken with a DOX aqueous solution [25]. However, we observed precipitation between ICG and DOX, leading us to adopt the active loading method [15]. Encapsulating chemotherapy drugs in liposomes has several advantages over free drugs, such as increased stability, reduced side effects, and precise controlled release. Moreover, the drug concentration can be conveniently adjusted using centrifuge tubes with selected filter pore sizes based on the diameter of the liposomes. As there is no universally applicable method for encapsulating all acidic, alkaline, hydrophobic, and hydrophilic drugs, finding the most suitable encapsulation method for each drug remains a challenge and an active area of research [24].

Our investigations have corroborated the previous findings and further expanded the conceptual framework into animal models through the application of a 780 nm laser. We have refined the liposome dimensions to 140 nm and have demonstrated enhanced theranostic effectiveness against lung cancer xenograft models utilizing a 780 nm laser. Additionally, the synthesis of an ICG-DOX nanocomposite through the conjugation of DOX with carbon dots has been validated in HepG2 xenograft models, demonstrating efficacy without therapeutic complications [16]. Our research conclusively indicates the presence of photothermal, photosensitizing, and ferroptotic effects within the Lipo-ICG/DOX framework, thereby enhancing the potential for clinical translation of this combination. In addition, our approach integrating a 780 nm laser system, higher ICG/DOX ratio, and ferroptosis-driven antitumor pathways offers an additional therapeutic advantage by efficiently combining imaging, drug delivery, and multi-mechanistic cancer cell eradication in one platform.

In the in vitro experiments, the results confirmed the great cell compatibility of ICG and the anticancer capability of doxorubicin based on the cell line model. Furthermore, the RNA-seq analysis results highlighted multiple pathways involved in the anticancer effects of Lipo-ICG/DOX on the A549 cells. The data suggests that the Lipo-ICG/DOX with light treatment group exhibited significant gene expression changes in pathways related to apoptosis, ferroptosis, and cell cycling. The marked reduction in anti-apoptotic genes such as BIRC5 and AKT1, alongside the increased expression of pro-apoptotic genes like DDIT3, GADD45A, and GADD45B, indicates that Lipo-ICG/DOX enhances apoptotic activity when combined with light treatment [26,27,28,29,30,31,32,33].

In the context of ferroptosis, the data revealed that the inhibition of ATF4 and the role of LMNB1 in inducing mitochondrial damage and ferroptosis are significant [28,29,34]. This aligns with the enhanced ferroptotic activity observed in the Lipo-ICG/DOX with light treatment group, further supporting its potential in inducing ferroptosis. Regarding cell cycle regulation, the analysis showed that Lipo-ICG/DOX with light treatment effectively inhibits cell proliferation by modulating key genes involved in cell cycle progression. The reduced expression of genes promoting cell cycle progression, such as E2F2, PLK1, CDC20, and CDC45, in the treatment group underscores the formulation’s ability to suppress cell proliferation, contributing to its overall anticancer efficacy [35,36,37,38,39].

The theranostic potential of our synthesized Lipo-ICG/DOX was further validated through an in vivo animal study. The discrepancy in laser wavelength selection across different studies (including ours) warrants close attention [16,17,19]. Notably, ICG in aqueous solution exhibits a principal absorption peak around 775–780 nm, particularly when bound to serum albumin, which can shift the dye’s nominal peak near 780 nm even further. At higher ICG concentrations, the transition from a monomeric to an oligomeric form can alter its optical properties [18]. Because 780 nm aligns more closely with ICG’s intrinsic absorption maximum—rather than 808 nm—it maximizes photon uptake and photothermal conversion within biological tissues, thereby enhancing treatment efficiency. Our in vivo xenograft model corroborates these findings, demonstrating robust antitumor efficacy when irradiated at 780 nm, underscoring the advantages of using a wavelength that more precisely matches ICG’s optimal absorption range.

In addition, the xenograft tumors were consistently visualized throughout the IVIS imaging periods, demonstrating the diagnosis capability provided by ICG and the effective enhanced permeability and retention (EPR) effect of the liposomal formulation. Liposomes accumulate in tumors due to the EPR effect and are metabolized by the liver, entering the digestive system through the bile duct without accumulating in the body. The complete metabolic process of our liposomes can be observed through the ICG signal. The prolonged duration of the ICG signal within the xenograft also indicates an extended light therapeutic window, which is crucial for clinical applications. Additionally, the xenografts exhibited the highest infrared signal intensity at 3–6 h post IVIS scanning, reducing damage to surrounding tissues caused by the light treatment of Lipo-ICG/DOX. Furthermore, our ex vivo IVIS images revealed that most infrared signals were located within the gastrointestinal (GI) tract rather than the heart, suggesting that the metabolism of Lipo-ICG/DOX primarily occurs through biliary excretion with partial renal excretion. This finding mitigates the risk of doxorubicin-induced cardiotoxicity and further underscores the distinct biodistribution profile of the liposomal formulation compared to the original anticancer drug.

## 5. Conclusions

We successfully prepared Lipo-ICG/DOX via an active loading method and verified its potent anticancer activity against A549 lung adenocarcinoma cells and their xenografts, both in vitro and in vivo. By leveraging ferroptotic and apoptotic pathways—along with imaging capabilities, photothermal effects, and controlled drug release—this multifunctional nanoformulation offers a promising and comprehensive approach to cancer therapy. In this research, we assessed only short-term in vivo antitumor efficacy rather than long-term outcomes, as the control group’s rapid tumor growth reached the experimental endpoint. A future study employing a single-arm design over an extended period could provide deeper insights into the long-term therapeutic benefits. Moving forward, efforts to refine dosing regimens, further delineate toxicity profiles, and explore repeated-dose protocols will be crucial to translating this platform into more durable and clinically feasible anticancer strategies.

## Figures and Tables

**Figure 1 pharmaceutics-17-00344-f001:**
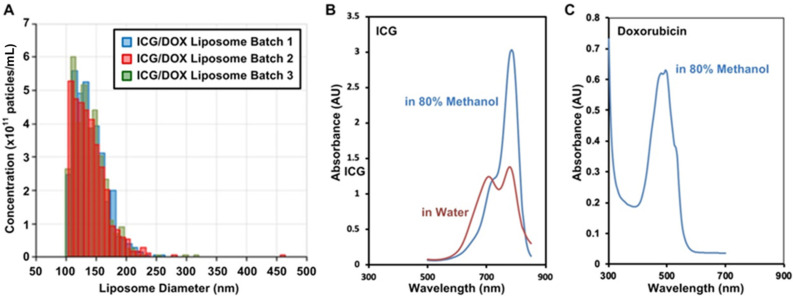
(**A**) Particle size distributions of the ICG/DOX liposomes fabricated in different batches. The size is analyzed using dynamic light scattering (DLS), and the average diameter of the liposomes is estimated to be approximately 140 nm. (**B**,**C**). The absorbance spectra of ICG and doxorubincin dissolved in different solutions.

**Figure 2 pharmaceutics-17-00344-f002:**
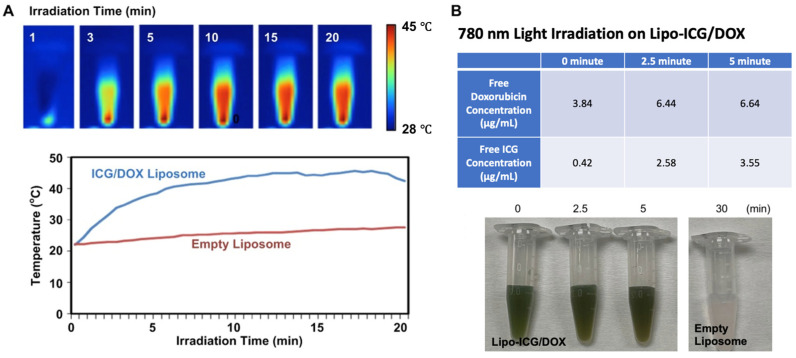
(**A**) Thermalgram images and the temperature variation in the solutions containing ICG/DOX liposomes irradiated with 780 nm light for 20 min to demonstrate the photothermal effect of the liposomes. The solution containing empty liposomes is used for comparison. (**B**) The table of free doxorubicin and ICG concentrations in the solution containing Lipo-ICG/DOX, and the experimental photos of the liposome solutions after exposure to 780 nm light for various periods.

**Figure 3 pharmaceutics-17-00344-f003:**
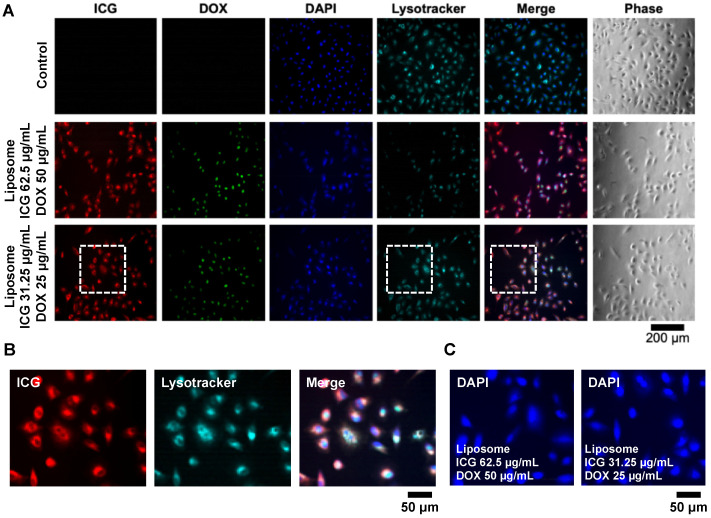
Comparison of the cellular uptake (A549 cells) of the ICG/DOX liposomes with different concentrations prepared in the experiments. The control group indicates the A549 cells without any liposome treatments. The cells are treated with the liposomes for four hours, and the liposomes are washed using DPBS and then fixed and stained. (**A**) The fluorescence and phase images of the cells treated with various conditions. (**B**) Zoom-in fluorescence images of the areas indicated by the white dash-line squares. The images show the co-localization of ICG and LysoTracker. (**C**) Zoom-in fluorescence images showing the nucleus of the cells treated with liposomes with different concentrations.

**Figure 4 pharmaceutics-17-00344-f004:**
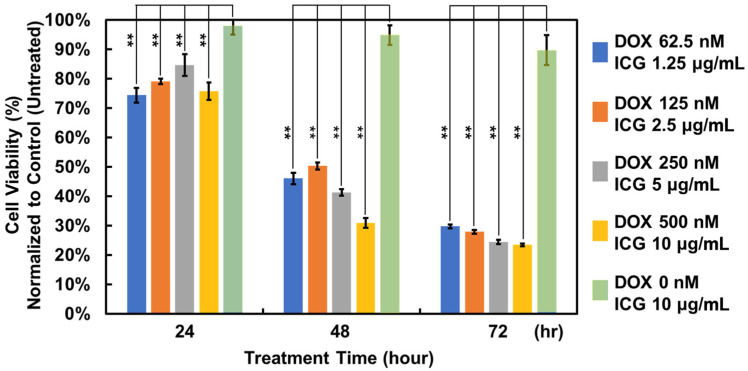
The cell viability of A549 cells treated with ICG/DOX liposomes with different concentrations for 24, 28, and 72 h. **, *p* < 0.01 represent significant differences according to Student’s *t*-test, comparing data from lipo-ICG/DOX treatment groups with lipo-ICG treatment group in the same assay conditions.

**Figure 5 pharmaceutics-17-00344-f005:**
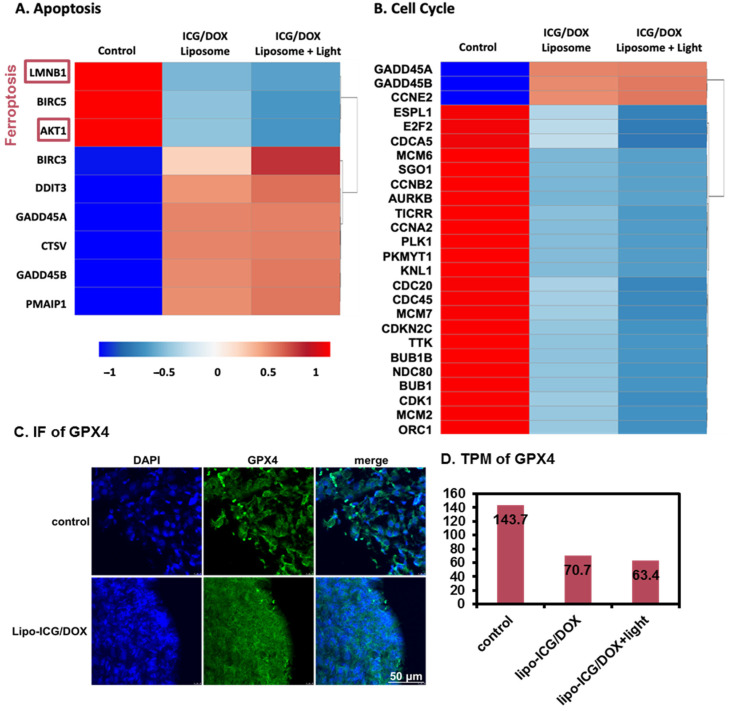
RNA-seq results showing the distinct gene patterns among the experiments performed without treatment (control) and with the ICG/DOX liposomes without and with the 20-min 780 nm light irradiation in aspects of (**A**) apoptosis, ferroptosis, and (**B**) cell cycle. (**C**) Immunofluorescence and (**D**) TPM from RNA-seq of GPX4.

**Figure 6 pharmaceutics-17-00344-f006:**
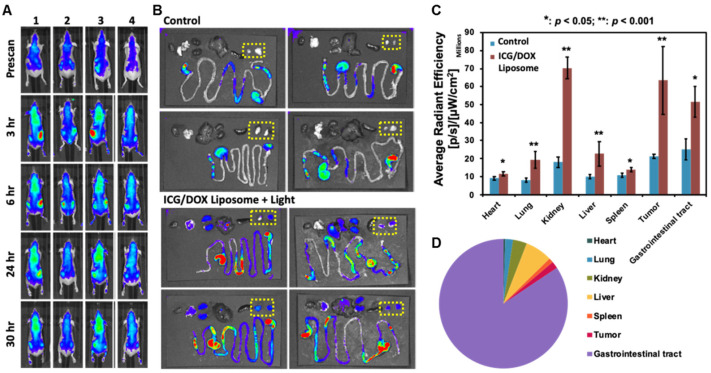
(**A**) IVIS images of the nude mice and their organs treated without and with Lipo-ICG/DOX treatment. The highest signal intensity areas were on the tumor xenograft 3–6 h of Lipo-ICG/DOX treatment. (**B**) The radiant efficiency of various organs from the mice treated without the liposomes (control) and the ICG/DOX liposomes. The tumor xenograft still exhibited near-infrared signal, whereas heart only showed marginal enhancement as compared with control group. (**C**) The bar chart further illustrated the degree of near-infrared enhancement of each organ. The kidneys, tumor, and GI tract show two-fold higher signal intensity enhancement. (**D**) The total radiant efficiency of various organs from the mice of ICG/DOX liposomes treatment groups were further illustrated in pie chart. The GI tract is responsible for 80% of the near-infrared signal.

**Figure 7 pharmaceutics-17-00344-f007:**
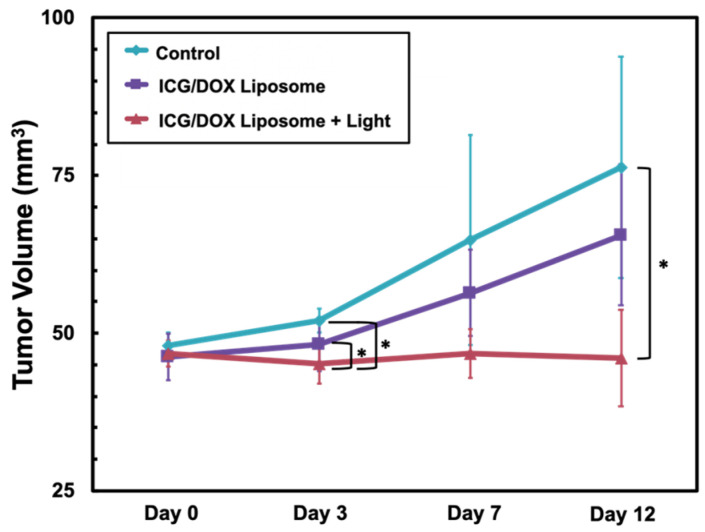
Plot of tumor growth in the nude mice without any treatment (control) and the mice treated with the ICG/DOX liposomes without and with the light irradiation. Tumor length (*L*) and width (*W*) are measured using a caliper, and the volume (*V*) is calculated by *V* = 0.5 × *L* × *W*^2^. *, *p* < 0.05 is significant differences according to Student’s *t*-test, comparing data from lipo-ICG/DOX treatment groups with control group in the same assay conditions.

## Data Availability

The original contributions presented in this study are included in the article/Appendix A. Further inquiries can be directed to the corresponding authors.

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
