# Peer review of "Multifaceted Functional Liposomes: Theranostic Potential of Liposomal Indocyanine Green and Doxorubicin for Enhanced Anticancer Efficacy and Imaging"

_pharmaceutics, 2025, doi:10.3390/pharmaceutics17030344_

Round 1
Reviewer 1 Report
Comments and Suggestions for Authors
The manuscript entitled "Multifaceted Functional Liposomes: Theranostic Potential of Liposomal Indocyanine Green and Doxorubicin for Enhanced Anticancer Efficacy and Imaging” by Wei-Ting Liao, Dao-Ming Chang, Meng-Xian Lin, Te-Sen Chou, Yi-Chung Tung and Jong-Kai Hsiao informs about the synthesis of liposomes containing Indocyanine Green and Doxorubicin and study of their dark and photoactivity toward lung adenocarcinoma in vitro and in vivo and their efficiency in imaging. The principal aim of this study was to demonstrate the theranostic potential of these liposomes. Authors have demonstrated the mechanism of liposomes includes contribution of ferroptosis and apoptosis.
Тhe results presented are of interest and the manuscript deserves publication in the journal Pharmaceutics after major revision.
Comments:
1. Authors consider the contribution of photothermic and photodynamic activity of Indocyanine Green in the mechanism of activity of mixed liposomes. However, unique result, which demonstrates the effect of light in the liposome activity, is the inhibition of tumour growth (Fig. 7). The authors need to clarify this aspect.
2. The Figure 4 shows that the principal liposome activity is associated with Doxorubicin and the role of Indocyanine Green is not clear. It is interesting to demonstrate its dark and photoactivity in the absence of Doxorubicin.
3. The text of the manuscript is quite difficult to read.
4. Тhe conclusion of the manuscript is too laconic.5. The English language of the manuscript must be corrected.
Comments on the Quality of English LanguageThe English language of the manuscript must be corrected.
Reviewer 2 Report
Comments and Suggestions for Authors
The topic of this work deals with the valorization of liposomes containing a near-infrared imaging agent (ICG), and an anti-cancer agent (DOX) as theranostic and therapeutic nanoformulations against lung adenocarcinoma. This work is very interesting, it shows in vitro and in vivo tests, as well as a pertinent RNA-seq analysis to better understand their action mechanism. Nevertheless, this article requires some improvements to be published in Phamaceutics. According to me, the main improvements concerns the quality of images presented in the Figures which has to be increased, and the discussion about methodology (i.e., active loading) and some results which has be more detailed.
Please, find my below suggested corrections.
** “Materials and Methods” part
Some experimental parameters are missing, they should be added:
- Line 115: What is the initial pH of 120 µM ammonium sulfate solution, and the final pH of the liposome formulation?
- Line 117: What is the temperature used during the extrusion step?
- Line 121: What is the MWCO of dialysis tubing?
- Lines from 122 to 126: The principle explaining “the replacement of ammonium sulfate with DOX” should be more explained (for instance, with a support of references).
- Line 144: The title “Liposome Treatment” is not explicit enough.
- Line 171: What is the duration of the irradiation at 780 nm at a power of 800 mW?
- Line 176: What is the temperature of centrifugation step at 4000 g for 15 minutes?
** “Results” part
Line 254: Concerning “the liposomes appeared as a turbid solution”, even after the extrusion step? This is surprising. The suspension of 140 nm-liposomes should be limpid.
Line 255: The issue of co-precipitation of ICG and DOX, mentioned in the “Discussion” part (lines 425-426), should be also presented here according to me.
Line 252: About “Liposomal encapsulation of ICG and DOX”, the word “characterization” should be removed because it makes unclear the title.
Line 258: The PDI value of liposomes, measured by DLS, should be added too. It gives some information about the size distribution of liposomes.
Line 262: Contrary to what is written, Figure 1B does not show the ICG solution in 80% methanol (but in water or methanol). This should be rewritten.
Line 263: The methodology of concentration determination by absorbance measurements should be added (use of calibration curves? concentration range?).
Line 278: What is the volume of liposomes put in the Eppendorf? This should be added.
Line 280: “…about 20 degrees after 60 minute irradiation” should be replaced by “…about 20 degrees after 20 minute irradiation”
Line 285: It would be interesting of adding 1 or 2 sentence(s) about the expected mechanism of ICG release under the irradiation treatment.
Line 301: The visualization of cells using an inverted fluorescence microscope in not shown in Figure 2, but in Figure 3.
Lines 301-302: It is very difficult on the Figure 3 to see the co-localization of the ICG and Lyso-Tracker signals. Is it possible to give enlargements?
This is exactly the same remark about the sentence “Additionally, it was observed…on nuclear staining” (lines 304-306)
Legend of Figure 3: The nature of the “control” should be written.
Figure 4: The color difference between DOX 62.5 nM and DOX 0 nM is not high enough.
Line 350: “(The concentrations are…)” should be replaced by “(the concentrations are…)”.
Line 369: “In the Immunofluorescence” should be replaced by “In the immunofluorescence”.
Line 371: For here and everywhere (mainly in the legends of Figures), the duration of the 780 nm light exposure should be indicated.
Figure 5C: The quality is not high enough, and the result is not explained enough too. Moreover, what is the dark area at right? Where is the immunofluorescence of liposomes-ICG/DOX after the irradiation?
Lines 386-389: According to me, these results are not at all supported by the Figure 6B which is not clear.
Line 392: How to explain that the tumor signal intensity is higher than the liver or spleen ones? Is it due to EPR as mentioned line 499? If yes, I think that a sentence about this could be also added at this step of text because this result is prominent.
Figure 6C and 6D: I do not understand the difference of representation between the both? Indeed, in Figure 6D, we show that the gastrointestinal tract signal corresponds to ca 80%, whereas it is not so high in Figure 6C. Some sentences explaining this could be important.
** “Discussion” part
Line 426: about “leading us to adopt the active loading method”, this could be developed (how?, with supporting references, and why? since in the introduction, it is written that ICG “can be dissolved in both aqueous and oil phases”.
It seems to me that some below bibliographic paragraphs could be better placed in the introduction in order to make more fluid this “Discussion” part:
- Lines 442-444
- Lines 448-450
- Lines 458-468
Moreover, the “sonodynamic effect”, mentioned by the authors as previously result (line 455, 462, 470), is confusing in this discussion which should be focused on the results presented in this paper.
According to me, the paragraph from line 511 to 521 is not clear, and misplaced (at the end of this discussion part).
In conclusion, this discussion part should concern more the results of considered work herein by presenting the main results and the associated assumptions.
Finally, “Limitations” and “Conclusion” parts should be merged to be a “Conclusion” part summarizing the main results and perspectives of this work (including the limitations).
** “Acknowledgements” and “Conflicts of interest” parts seem to be inverted.
Reviewer 3 Report
Comments and Suggestions for Authors
The authors synthesized ICG and DOX loaded liposomes and evaluated their theranostic potential in vitro and in vivo using lung adenocarcinoma models. The topic area falls within the scope of the Journal of Pharmaceutics, but needs additional experiments.
1. Please separate chemicals and materials list from method/protocols session.
2. Explain how the unloaded drug (ICG and Dox) was purified on page 3
3. The author must include cell viability test for serial concentration of free drugs, (ICG, DOX) or combination (ICG+Dox) with and without irradiation and ICG/DOX liposomes toxicity in the presence of laser irradiation like Figure 4.
4. The author must include ICG/Dox release, with and without laser irradiation from ICG/DOX liposomes.
5. The author used some of their figures (Figure 2A and 2B) from previously published paper (Figure 2B and 2C), attached link (https://pmc.ncbi.nlm.nih.gov/articles/PMC10891763/pdf/pharmaceutics-16-00224). They manipulated time from second to minutes which is plagiarism
6. Discuss more in detail on the RNA-seq data
Comments on the Quality of English Languagecan be improved
Round 2
Reviewer 1 Report
Comments and Suggestions for Authors
I am satisfied with the corrections and answers and believe that the manuscript can be published in its current form.
Author Response
Thanks for the reviewer's support.
Reviewer 2 Report
Comments and Suggestions for Authors
This paper can be accepted in present form.
Author Response
Thanks for the reviewer's support.
Reviewer 3 Report
Comments and Suggestions for Authors
The authors have addressed all the essential comments, and the revised manuscript is suitable for publication.
Author Response
Thanks for the reviewer's support.